# Loss of 5′-Methylthioadenosine Phosphorylase (MTAP) is Frequent in High-Grade Gliomas; Nevertheless, it is Not Associated with Higher Tumor Aggressiveness

**DOI:** 10.3390/cells9020492

**Published:** 2020-02-20

**Authors:** Weder Pereira de Menezes, Viviane Aline Oliveira Silva, Izabela Natália Faria Gomes, Marcela Nunes Rosa, Maria Luisa Corcoll Spina, Adriana Cruvinel Carloni, Ana Laura Vieira Alves, Matias Melendez, Gisele Caravina Almeida, Luciane Sussuchi da Silva, Carlos Clara, Isabela Werneck da Cunha, Glaucia Noeli Maroso Hajj, Chris Jones, Lucas Tadeu Bidinotto, Rui Manuel Reis

**Affiliations:** 1Molecular Oncology Research Center, Barretos Cancer Hospital, Barretos, São Paulo 14.784-400, Brazil; wedermenezes1@gmail.com (W.P.d.M.); vivianeaos@gmail.com (V.A.O.S.); izabela.faria.tk@hotmail.com (I.N.F.G.); nr.marcela2@gmail.com (M.N.R.); mluisaspina@hotmail.com (M.L.C.S.); drybiomedic@gmail.com (A.C.C.); alves.anav@gmail.com (A.L.V.A.); matiasmelendez@gmail.com (M.M.); lsussuchi@gmail.com (L.S.d.S.); lucasbidinotto@gmail.com (L.T.B.); 2Department of Pathology, Barretos Cancer Hospital, Barretos, São Paulo 14.784-400, Brazil; giselecaravina@gmail.com; 3Department of Neurosurgery, Barretos Cancer Hospital, Barretos, São Paulo 14.784-400, Brazil; carlosclara.neuro@gmail.com; 4A.C Camargo Cancer Center, São Paulo, São Paulo 015.080-10, Brazil; isabela.werneck@rededor.com.br (I.W.d.C.); ghajj@cipe.accamargo.org.br (G.N.M.H.); 5Institute of Cancer Research, London SW7 3RP, UK; chris.Jones@icr.ac.uk; 6Barretos School of Health Sciences, Dr. Paulo Prata - FACISB, Barretos, São Paulo 14.785-002, Brazil; 7Department of Pathology, Botucatu Medical School, Universidade Estadual Paulista – Unesp, Botucatu, São Paulo 18.618-970, Brazil; 8Life and Health Sciences Research Institute (ICVS), School of Medicine, University of Minho, 4710-057 Braga, Portugal; 93B’s - PT Government Associate Laboratory, 4806-909 Braga/Guimarães, Portugal

**Keywords:** glioma, glioblastoma, 5’-methylthioadenosine phosphorylase (MTAP), immunohistochemistry, tumor biology, proliferation, migration, invasion

## Abstract

The 5’-methylthioadenosine phosphorylase (MTAP) gene is located in the chromosomal region 9p21. *MTAP* deletion is a frequent event in a wide variety of human cancers; however, its biological role in tumorigenesis remains unclear. The purpose of this study was to characterize the MTAP expression profile in a series of gliomas and to associate it with patients’ clinicopathological features. Moreover, we sought to evaluate, through glioma gene-edited cell lines, the biological impact of MTAP in gliomas. MTAP expression was evaluated in 507 glioma patients by immunohistochemistry (IHC), and the expression levels were associated with patients’ clinicopathological features. Furthermore, an in silico study was undertaken using genomic databases totalizing 350 samples. In glioma cell lines, MTAP was edited, and following MTAP overexpression and knockout (KO), a transcriptome analysis was performed by NanoString Pan-Cancer Pathways panel. Moreover, MTAP’s role in glioma cell proliferation, migration, and invasion was evaluated. Homozygous deletion of 9p21 locus was associated with a reduction of *MTAP* mRNA expression in the TCGA (The Cancer Genome Atlas) - glioblastoma dataset (*p* < 0.01). In addition, the loss of *MTAP* expression was markedly high in high-grade gliomas (46.6% of cases) determined by IHC and Western blotting (40% of evaluated cell lines). Reduced *MTAP* expression was associated with a better prognostic in the adult glioblastoma dataset (*p* < 0.001). Nine genes associated with five pathways were differentially expressed in MTAP-knockout (KO) cells, with six upregulated and three downregulated in MTAP. Analysis of cell proliferation, migration, and invasion did not show any significant differences between MTAP gene-edited and control cells. Our results integrating data from patients as well as in silico and in vitro models provide evidence towards the lack of strong biological importance of MTAP in gliomas. Despite the frequent loss of MTAP, it seems not to have a clinical impact in survival and does not act as a canonic tumor suppressor gene in gliomas.

## 1. Introduction

Gliomas represent the most common group of primary central nervous system (CNS) tumors [1]. According to the World Health Organization (WHO), these tumors can be categorized in low-grade gliomas (LGG, WHO grades I and II) and high-grade gliomas (HGG, WHO grades III and IV) according to their histological and molecular features [2,3]. Pilocytic astrocytoma (PA) (WHO grade I) is a good-prognosis childhood tumor, and conversely, glioblastoma (GBM) (WHO grade IV) occurs mainly in adults and has the poorest prognosis [2,4]. GBM has a mean survival of ~14 months after gold-standard therapy—surgery—followed by radiation therapy plus concomitant temozolomide [5,6,7]. GBM accounts for 70% of gliomas and can be subdivided in GBM *IDH*-wild type (the most frequent, >90%), previously known as primary (de novo) GBM and exhibiting a short clinical history, and GBM *IDH*-mutant, also called secondary GBM, which results from the malignant progression from lower-grade gliomas of diffuse (WHO grade II) or anaplastic (WHO grade III) astrocytomas [6,8] and is related to point mutations in IDH1/2 genes. 

In fact, several efforts have been made to improve the molecular characterization of GBM. In 2014, the International Society of Neuropathology recommended the support of molecular analysis for determining brain tumor entities [9] and, in 2016, the World Health Organization proposed the use of molecular features, including *IDH* mutation to GBM, in addition to histologic features in the tumor entities [3]. Additionally, there are many other biomarkers studied in GBMs as subjects of special attention. *TERT* promoter mutations have been found to be markedly high in primary GBMs (from 54% to 83% of the cases) [10]. Further studies have shown poorer survival of *TERT*-mutated patients in comparison to *TERT*-wild type [11,12,13]. Finally, studies have found an interaction between *TERT* promoter mutation and *MGMT* methylation. By performing pairwise comparisons, it was identified that *MGMT* methylation improved the survival of *TERT* promoter mutated-patients [14]. On the other hand, *TERT*-mutant and *MGMT* unmethylated patients presented the poorest prognosis, pointing to a possible impact in the use of *IDH*, *TERT,* and *MGMT* in the improvement of diffuse gliomas classification and prognostication [15].

We previously described the most frequent chromosomal alterations in a series of Brazilian astrocytomas [11]. We identified chromosome 7 gain, *EGFR* amplification, and losses in chromosomes 9p, 10, and 13, in accordance with other populations [4,11,16]. We also found 9p- deletion in approximately 50% of GBMs, affecting primarily the 9p21 locus where several tumor suppressor genes are located, including *CDKN2A/B* and *MTAP* [11].

*MTAP* (5′-methyltioadenosine phosphorylase) encodes a key enzyme involved in the metabolism of polyamines and purines [17,18,19]. This enzyme converts 5′-methyltioadenosine (MTA), a by-product of polyamine biosynthesis, into adenine and MTR-1-P (methylthioribose-1-phosphate), which are recycled into AMP (adenosine monophosphate) and methionine [19,20]. This protein is expressed virtually in all tissues throughout the body, and its homozygous deletion is frequently associated with solid and hematologic tumors such as mesothelioma, lung carcinoma, hepatocellular carcinoma, gastrointestinal stromal tumors, metastatic melanoma, leukemias, and lymphoma [18,21,22,23,24,25]. Therefore, *MTAP* has been reported as a tumor suppressor gene [24,26,27,28,29,30]; however, many studies have demonstrated the contradictory function of *MTAP*. For instance, the loss of MTAP expression has been associated with inhibition of growth and progression of head and neck carcinoma and lung cancer by MTA accumulation [31,32]. Bistulfi et al. [33] showed that knockdown of MTAP blocks prostate cancer growth in vitro and *in vivo*. In addition, methionine deprivation acts by inhibiting cell migration, invasion, and metastasis in breast cancer [34]. Metabolic changes in tumors, especially those relating to polyamines metabolism, demonstrate that many mechanisms underlying MTAP function need to still be clarified [35,36]. In gliomas, loss of *MTAP* locus is also frequently reported [37,38,39,40,41]. Nevertheless, the clinical and the biological impacts of MTAP are poorly explored in gliomas [37,38,42]. 

Therefore, the aim of this study was to characterize the MTAP protein expression profile in a large series of glioma and to associate it with the patients’ clinicopathological features. Moreover, by using glioma cell lines, the biological role of *MTAP* was evaluated. By integrating data from patients and in vitro models, this study showed that, despite the frequent loss of *MTAP*, it does not have a clinical impact in survival and does not act as a canonic tumor suppressor gene in gliomas.

## 2. Materials and Methods

### 2.1. Cell Lines and Gene Editing

One cell line derived from normal astrocytes (NHA), seven short-term primary glioma cell lines, and 11 established glioma cell lines were evaluated. The MTAP positive U251 cell line was transfected with MTAP Clustered Regularly Interspaced Short Palindromic Repeats (CRISPR)/CRISPR-associated protein 9 (Cas9) (MTAP-CRISPR/Cas9 KO) or empty vector CRISPR/Cas9 Plasmid (Santa Cruz Biotechnology, Santa Cruz, CA, USA), leading to a U251 MTAP−/− clone. The MTAP negative SW1088 cell line was transduced with the MTAP human lentivirus or the blank control lentiviral vector in accordance with the manufacturer’s instructions (ABM Inc.^®^, Richmond, BC, Canada), resulting in SW1088 MTAP+/+ clones. Detailed information about the cell lines used and the gene editing are described in the Appendix B material.

### 2.2. Patients

Adult and pediatric glioma tissues were obtained from 507 patients who underwent surgery for glioma at Barretos Cancer Hospital (BCH), Hospital of Clinics of Faculty of Medicine of São Paulo University (HCRP), AC Camargo Cancer Center, and The Institute of Cancer Research between 1980 and 2013. Histologic review of the slides was performed by expert neuropathologists (according to the latest WHO histopathological criteria) [3] to confirm the diagnosis. Overall, patient age ranged from 0.3 to 82.8 years (median: 57 years old). The stratification showed 49 (9.7%) patients in pediatric (0–19 years) and 458 (90.3%) in the adult group (>19 years). Histological subtypes of gliomas were distributed into diffuse astrocytoma (n = 18), anaplastic astrocytoma (n = 24), pediatric glioblastoma (n = 42), and adult glioblastoma (n = 423). This study was approved by the Ethics Committee of Barretos Cancer Hospital under approval number 630/2012, number 1175879 (961/2015), and AC Camargo Cancer Center (number 1485/10).

### 2.3. DNA Isolation

Tissue from a patient’s tumor was manually microdissected from 4 μm unstained histological sections. DNA was isolated from each target using the DNeasy Blood and Tissue kit (Qiagen, Valencia, CA, USA) according to the protocols provided by the supplier. The 260/280 and the 260/230 ratios were determined by a NanoDrop 2000C spectrophotometer (Thermo Scientific, Wilmington, DE, USA), and the DNA was quantified using Quant-iT PicoGreen dsDNA (Invitrogen, Eugene, OR, USA) according to the supplier’s protocol.

### 2.4. RNA Extraction and RT-qPCR

RNA from cultured cell lines was isolated using a modified TRIzol^®^ reagent protocol (Thermo Scientific, Waltham, MA, USA) [43]. In brief, cultured cells were washed twice with Dulbecco’s phosphate-buffered saline (DPBS; Thermo Scientific, Waltham, MA, USA), and 1.5 mL TRIzol^®^ was added. Flasks were scraped, and TRizol/cell mixture was transferred to a 1.5 mL eppendorf tube. Subsequently, 200 µL of chloroform was added, homogenized for 30 s, and centrifuged at 17,982× *g* for 15 min. The supernatant was collected, and 750 µL isopropanol was added and kept overnight at −20 °C. Samples were then centrifuged at 17,982× *g* for 10 min at 4 °C. The resulting RNA pellet was washed twice with 75% ethanol after removing the supernatant. Finally, the RNA pellet was dried and dissolved in 20 µL of ultra-pure water. The integrity of all the RNA preparations was checked by RNA 600 nano assay (Agilent Technologies, Santa Clara, CA, USA), and RNA concentrations were measured with NanoDrop™ 2000C spectrophotometer (Thermo Scientific, Waltham, MA, USA). Complementary DNA (cDNA) was synthesized using Superscript III reverse transcriptase (Thermo Scientific, Waltham, MA, USA) according to the supplier protocol.

RT-qPCR reactions were carried out in a total volume of 10 µL using 2X Fast SYBR® Green Master Mix kit (Thermo Scientific, Waltham, MA, USA) containing 100 ng (1 µL) of cDNA, 10 µM (0.8 µL) of each primer, and 7.4 µL of ultra-pure water in a final volume of 20 µL. Gene amplification was performed with the programmable cyclic reactor StepOne ™ Real-Time PCR System (Applied Biosystems, Grand Island, USA) as follows: 95 °C for 10 min; 40 cycles at 95 °C for 15 s; and 64 °C for 1 min. The primers used for RT-qPCR amplification of *MTAP* were forward primer: 5′- TCTTGTGCCAGAGGAGTGTG-3′; reverse primer: 5′-ACCATTGTCCCCTTTGAGTG-3′. Samples were then normalized using the housekeeping gene *HPRT1* (forward primer: 5′-GACCAGTCAACAGGGGACAT-3′; reverse primer: 5′-CTGCATTGTTTTGCCAGTGT-3′). The normalized expressions of the gene of interest were calculated using the 2^-ΔCt^ method. The primers used were synthesized by Sigma-Aldrich, St. Louis, MO, USA, and all reactions were performed in three biological replicates.

### 2.5. IDH1 Mutation Analysis

The analysis of hotspot mutations of isocitrate dehydrogenase 1 (IDH1-exon 4) was performed by PCR followed by direct sequencing. Briefly, the IDH1 region of interest was amplified by PCR using the following primers: 59-CGGTCTTCAGAGAAGCCATT-39 (forward) and 59-CACATTATTGCCAACATGAC-39 (reverse) [11]. An amplification PCR reaction was performed in a total volume of 15 µL comprising: 1 µL of DNA, 1X buffer solution, 2 mM MgCl2, 200 µM of each dNTP, 0.3 µM of each primer set, and 0.5 U of *Taq* DNA polymerase (Invitrogen, Eugene, OR, USA), and was performed in a Veriti 96-well Thermal Cycler with an initial denaturation at 95 °C for 10 min, amplification for 40 cycles with denaturation at 95 °C for 45 s, annealing at 58 °C for 45 s, extension at 72 °C for 45 s, and a final extension at 72 °C for 10 min. Amplification of PCR products was confirmed by gel electrophoresis. PCR sequencing was performed using the Big Dye terminator v3.1 cycle sequencing ready reaction kit (Applied Biosystems, Grand Island, NY, USA) and an ABI PRISM 3500 xL Genetic Analyzer (Applied Biosystems, Grand Island, NY, USA).

### 2.6. TERT Mutation Analysis

The analysis of hotspot mutations in the promoter region of the telomerase reverse transcriptase gene (*TERT*) was performed by PCR followed by direct Sanger sequencing [44]. Briefly, the *TERT* promoter region was amplified by PCR using the following primers: 59-AGTGGATTCGCGGGCACAGA-39 (forward) and 59-CAGCGCTGCCTGAAACTC-39 (reverse), leading to a 235 bp PCR product containing C228T and C250T mutations. Amplification by PCR was performed with an initial denaturation at 95 °C for 15 min, followed by 40 cycles of denaturation at 95 °C for 30 s, annealing at 64 °C for 90 s, elongation at 72 °C for 30 s, and final elongation at 72 °C for 7 min. Amplification of PCR products was confirmed by gel electrophoresis. PCR sequencing was performed using the Big Dye terminator v3.1 cycle sequencing ready reaction kit (Applied Biosystems, Grand Island, NY, USA) and an ABI PRISM 3500 xL Genetic Analyzer (Applied Biosystems, Grand Island, NY, USA).

### 2.7. MGMT Promoter Methylation

To evaluate the methylation status of the promoter region of the O^6^-methylguanine-DNA methyltransferase gene (*MGMT*), sodium bisulfite treatment of DNA (300–1500 ng) was performed using the Epitect Kit (Qiagen, Valencia, CA, USA) according to the manufacturer’s instructions. Methylation-specific (MS) PCR for the MGMT promoter was performed as described previously [45].

### 2.8. Western Blotting Analysis

The Western blot analysis was performed as previously described [42]. In brief, cells were lysed with lysis buffer (50 mM Tris; pH 7.6; 150 mM NaCl; 5 mM EDTA; 1 mM Na_3_VO_4_; 10 mM NaF; 10 mM sodium pyrophosphate; 1% NP-40) containing protease inhibitor cocktail (0.01 M EDTA; 1 mM DTT; 1 mM Leupeptin; 1 mM PMSF; and 1 µM Aprotinin). The cell lysate was placed on ice for 60 min and centrifuged at 17,982× *g* for 15 min. The total protein was quantified using the Bradford method (Quick Start ^TM^ Bradford Protein Assay; Bio-Rad, Hercules, CA, USA). Total proteins were separated by 12% polyacrylamide gel and transferred onto Trans-Blot Turbo Midi Nitrocellulose Transfer Packs (Bio-Rad, Hercules, CA, USA). The membrane was incubated with primary followed by secondary antibodies after blocking with 5% non-fat milk. Immunodetection was performed using the ECL KIT (Amersham Biosciences, Uppsala, Uppland, Sweden) in ImageQuant LAS 4000 mini (GE Healthcare Life Sciences, Pittsburgh, PA, USA) and quantified by the platform for scientific image analysis ImageJ (NIH) [46]. The following antibodies were used: polyclonal anti-MTAP antibody (Proteintech, Rosemont, IL, USA) diluted 1:800 and anti-β actin antibody (Cell Signaling Technology, Danvers, MA, USA) diluted at 1:2000.

### 2.9. Immunohistochemistry Analysis

For the immunohistochemistry (IHC) analysis, a tissue microarray (TMA) was constructed with tumors areas obtained from each case and inserted in the recipient paraffin block using the tissue arrayer MTA-1 platform (Beecher Instruments^TM^, Silver Springs, MD, USA). To represent possible heterogeneity of tumors, two 1.0 mm cores from each case were used in TMA blocks. IHC was performed from 4.0 µm sections of the TMA block for MTAP, as previously described [42]. Glass slides were deparaffinized and subjected to antigen retrieval in a Pascal pressure chamber (Dako, Carpinteria, CA, USA). The primary MTAP polyclonal antibody (Proteintech, Rosemont, IL, USA) was diluted 1:300 with the background-reducing medium and kept at 4 °C overnight. The Avidin–Biotin Complex (ABC) method was performed according to the manufacturer`s recommendations with staining with DAB (3,3’ Diaminobenzidine) and counterstaining with Harris’ hematoxylin (Leica Biosystems, Newcastle, UK). The extension of immunoreactions was measured according to the following criteria: 0 = negative; 1 = ≤25% of positive cells; 2 = 25–50% of positive cells; 3 = ≥50% of positive cells. Moreover, the intensity of reaction was defined as: 0= negative; 1 = weak; 2= moderate; 3 = strong. Cases presenting the sum of the scores (extension and intensity) between 0 and 3 were considered negative, and those presenting scores between 4 and 6 were considered positive. Non-neoplastic tissue microarray (BN 961-Biomax INC, Rockville, MD, USA) was used as a positive control for IHC reactions.

### 2.10. In Silico Analysis

*MTAP* copy number alterations (CNA) data were downloaded from comparative genomic hybridization/single-nucleotide polymorphism (CGH_SNP) arrays (n = 350) [47]. Based on normalized *MTAP* values, patients were stratified in normal (n = 150) (>−0.1) and *MTAP* homozygous deletion carriers (n = 200) (≤−1.5). For expression analysis, *MTAP* mRNA G450 array data were downloaded (n = 299) [47]. The median *MTAP* value was calculated considering all samples. Patients with *MTAP* values at least 20% higher than the median were considered positive (n = 138), while those with *MTAP* values at least 20% lower than the median were considered negative (n = 161). Finally, for methylation analysis, *MTAP* methylation beta values were downloaded from 283 samples [47]. Patients were stratified as hypermethylated (beta > 0.5) and hypomethylated (beta ≤ 0.5). Integrated in silico analyses of MTAP CNA expression and methylation status were performed using the TCGA2STAT package implemented in R software downloaded from https://www.r-project.org/. Moreover, in silico analysis was conducted to evaluate *MTAP* expression in glioblastoma subtypes in The Cancer Genome Atlas (TCGA) dataset. The median *MTAP* value was calculated considering all samples.

### 2.11. mRNA NanoString^TM^ Data Analysis

Gene expression analysis on *MTAP* gene-edited U251 and SW1088 cell lines was performed using the NanoString nCounter PanCancer Pathways panel (730 gene transcripts distributed in 13 biological pathways) according to the manufacturer’s standards (NanoString Technologies, Seattle, WA, USA). Moreover, 24 of 730 genes associated with tumor aggressiveness were selected, and the differential expression was evaluated to better understand the function of *MTAP* gene-edited on migration and invasion ability. Briefly, 100 ng aliquots of RNA were hybridized with probe pools, hybridization buffer, and TagSet reagents in a total volume of 30 μL and incubated at 65 °C for 20 h. After codeset hybridization overnight, the samples were washed and immobilized to a cartridge using the Nanostring nCounter Prep Station (NanoString Technologies, Seattle, WA, USA) for 4 h. Finally, the cartridges containing immobilized and aligned reporter complexes were scanned in the nCounter Digital Analyzer (NanoString Technologies, Seattle, WA, USA), and image data were subsequently generated using the high-resolution setting, which takes 577 images per sample. Quality control assessment of raw NanoString gene expression counts was performed with nSolver Analysis Software version 2.5 and the default settings (NanoString Technologies, Seattle, WA, USA). Normalization with internal positive controls and housekeeping genes was performed in R statistical environment using NanostringNorm package [48]. Normalized log2 mRNA expression values were used for subsequent data analysis. Genes with fold change (FC) ±2 and *p* < 0.05 were considered significant.

### 2.12. xCELLigence Proliferation Assay

U251 MTAP−/− and SW1088 MTAP+/+ cell proliferation was monitored with a real-time cell analyzer using xCELLigence Technology (Roche Applied Science, Indianapolis, IN, USA). This impedance value was measured by the Real Time Cell Analysis Dual purpose (RTCA DP) system and is reported in the dimensionless unit of cell index. Prior to seeding of cells in the 96-well E-plate, 100 µL Dulbecco’s modified Eagle’s medium (DMEM) was added to wells, and background was recorded. U251 MTAP−/−; U251 EV; U251 WT and SW1088 MTAP+/+; SW1088 LB; SW1088 WT were split, and 5–10 × 10^3^ cells in 100 µL of media supplemented with fetal bovine serum (FBS; 10%) were added to each well. Real-time monitoring of cell proliferation measured as cell index was recorded every 5 min for up to 72 h. Data represent changes in cell index over time.

### 2.13. Transwell Migration Assay

Boyden chamber type Transwell permeable supports (Corning, Chelmsford St. Lowell, MA, USA) were used to examine the ability of cells to invade the 8 μm pore size membrane (1 × 10^5^ pores/cm^2^) and migrate through the polyethylene terephthalate (PET)-membrane surface. The CRISPR-edited cell line (U251 MTAP−/−) and the stable expression cell line (SW1088 MTAP+/+) beside the control cell lines were seeded into the inner chamber at 5 × 10^4^ and 10 × 10^4^, respectively, in 500 µL of DMEM without serum and allowed to migrate onto the coated undersurface at 37 °C in a CO_2_ incubator for 24 h. At the bottom, 750 µL of DMEM supplemented with 10% FBS was added to the chambers. Migrated cells were fixed and stained with hematoxylin and eosin for three minutes followed by phosphate-buffered saline (PBS) washing of the membranes. The migrated cells in each well were counted in ten different fields per experiment under the microscope. The mean values were obtained from three replicate experiments and were subjected to t-test.

### 2.14. Cell Invasion Assay

U251 MTAP−/− and SW1088 MTAP+/+ beside control cells were cultured to near confluence in DMEM+10% FBS. Cells were harvested by trypsinization and washed and suspended in DMEM without serum at 2 × 10^5^ and 4 × 10^5^ cells/mL. Prior to preparing the cell suspension, the dried layer of matrigel matrix was rehydrated with serum-free DMEM for 1 h at 37 °C. The rehydration solution was carefully removed, 750 µL DMEM containing 10% FBS was added to each well as a chemoattractant, and 500 µL (1–2 × 10^5^ cells) of cell suspension was added into the inner chamber Corning, Chelmsford St. Lowell, MA, USA). The plates were incubated for 24 h at 37 °C. The invasive cells on the bottom surface of the membrane were fixed and stained with hematoxylin and eosin for three minutes followed by 1X PBS washing of the membranes. The migrated cells in each well were counted in ten different fields per experiment under the microscope. The mean values were obtained from three replicate experiments and were subjected to t-test.

### 2.15. Statistical Analysis

The statistical analysis was performed using SPSS version 24 for Windows ^TM^ (IBM, Chicago, Ill, USA) considering statistically significant values of *p* < 0.05. Levene and Kolmogorov–Smirnov tests were performed to test the variance and the normality of the data. One-way ANOVA or Kruskal–Wallis tests were performed to compare copy number, expression, and methylation of *MTAP* profiles. ANOVA followed by Tukey post hoc test was performed to evaluate *MTAP* expression in glioblastoma subtypes. The association between clinicopathological categorical variables and *MTAP* was analyzed by Pearson’s chi-square test or Fisher exact test. The overall survival (OS) analysis was done by constructing Kaplan–Meier curves, and the log-rank test was applied for the comparison between the obtained curves. OS was defined as the time between the date of diagnosis and the date of the patient’s last information or death. Cell line experiments were analyzed with Student’s t-test and are expressed as mean values of at least three independent experiments ± standard errors. Differences were considered significant at * *p* < 0.05, ** *p* < 0.01, and *** *p* < 0.001.

## 3. Results

### 3.1. Loss of MTAP Expression is Associated with 9p21 Locus Deletion in Gliomas

Initially, in order to understand the molecular mechanism underlying *MTAP* expression, we investigated the impact of copy number alteration (CNA) of 9p21 locus on *MTAP* mRNA expression levels in a TCGA-GBM dataset (n = 350). As shown in Figure 1A, *MTAP* gene expression was significantly (*p* < 0.01) lower in the homozygously deleted group (n = 169) than in those with normal ploidy (n = 181). Since *MTAP* regulation can also occur by epigenetic mechanisms [27], we then evaluated the association of *MTAP* mRNA expression and the methylation status in 283 samples from the TCGA-GBM dataset (Figure 1B). *MTAP* gene promoter was methylated in 60 of 283 (21.2%) samples. However, no association of methylation status and *MTAP* expression levels in glioblastomas was observed (Figure 1B, *p* = 0.633). Additionally, we evaluated the *MTAP* expression levels in the five molecular subtypes of TCGA-GBM (n = 291) (Table 1). We found loss of *MTAP* expression greater than 50% in almost all subtypes analyzed, except for the glioma CpG island methylator phenotype (G-CIMP) (Table 1). The classical subtype presented a higher frequency of loss of *MTAP* gene expression with 65.2% (43/66) against only 7.4% (2/25) for the G-CIMP+ subtype (*p* < 0.001) (Table 1).

Next, in a set of eleven established and seven short-term primary gliomas cells, we compared MTAP copy number with MTAP mRNA and protein expression by RT-qPCR and Western blot, respectively (Appendix A). Overall, we found loss of *MTAP* mRNA expression in 33.3% (6/18) of the cell lines evaluated (Figure 2), particularly in 36.3% (4/11) of established cell lines, with 25% (1/4) for pediatric and 42.8% (3/7) for adult glioma cell lines (Figure 2A), and 29% (2/7) of patient-derived glioma cell lines (Figure 2B). A 100% concordance was found between protein and mRNA levels (Figure 2C–F). Of note, 40% (6/15) of high-grade glioma (HGG; WHO grade III–IV) cell lines presented loss of *MTAP* expression, contrasting with 0% in low-grade glioma (LGG; WHO grade I–II) cell lines (0/3) (Figure 2A,B). Considering only the GBM subtype, a lack of *MTAP* mRNA and protein expression was found in 33.3% (4/12) of samples, with 44.4% (4/9) of adult GBM cell lines having no detectable MTAP in the cell extracts (Figure 2C,D). These results suggest that MTAP loss of expression is associated with higher-grade gliomas.

### 3.2. MTAP Expression Profile and Clinicopathological Association in Gliomas

We further evaluated MTAP protein expression by immunohistochemistry in a large set of diffuse gliomas (n = 507) (Table 2 and Figure 3). Overall, we found loss of *MTAP* expression in 45.95% (233/507) of cases, this being observed in 27.8% (5/18) of diffuse astrocytoma, 50.0% (12/24) of anaplastic astrocytoma, 45.6% (193/423) of adult glioblastoma, and 54.8% (23/42) of pediatric glioblastoma (Figure 3A–E). When we stratified the samples by grade, the loss of *MTAP* expression in the high-grade glioma subgroup was almost two-fold greater (46.6%, 228/489) than in the low-grade glioma subgroup (27.8%; 5/18) (Table 2 and Figure 3E). However, we did not observe a significant association between *MTAP* expression and other patients’ clinicopathological features, such as gender, age, or Karnofsky Performance Status (KPS), except for tumor location (*p =* 0.013) (Table 2).

Following the assessment of *TERT, IDH1* mutational profile, and *MGMT* methylation status in a subset of cases, the association of these molecular alterations with the loss of MTAP protein expression was evaluated. No significant association was observed (Table 2).

Importantly, we evaluated the impact of *MTAP* expression on overall survival (OS) of patients diagnosed with GBMs. In total, 398 adult patients were included in the OS analysis from which 83.1% (n = 331) died during total observation time: 76.6% (n = 138) in the MTAP-negative group and 88.5% (n = 193) in the MTAP-positive group (Figure 4A). We found that, in adult glioblastoma, patients exhibiting a lack of *MTAP* expression had better survival than those presenting *MTAP* expression (median survival of 9.8±0.86 vs. 6.23±0.70 months, respectively, *p* = 0.00023 in log-rank test) (Figure 4A). Two-year OS rates were 10.6% for the MTAP-negative group and 7.6% for the MTAP-positive group. In the pediatric context, 40 pediatric glioblastoma patients included in the analysis—86.3% (n = 19) in the MTAP-negative group and 77.7% (n = 14) in the MTAP-positive group—died during the total observational time (Figure 4B). There was no association of MTAP loss and patients’ OS, with a median OS of 13.0±1.71 months in the MTAP-negative group and 14.0±4.83 months for the MTAP-positive group (*p* = 0.499 in log-rank test) (Figure 4B). The two-year OS rates were 7.2% and 5.56% for MTAP-negative and MTAP-positive groups, respectively.

In order to extend these findings, we performed in silico survival analysis considering CNA status in TCGA-GBM datasets (n = 246). The analysis did not demonstrate a significant difference between *MTAP* CNA and overall survival (Figure 4C; *p =* 0.942). We further evaluated disease-free survival (DFS) considering relapse and death as endpoints. Similarly, the five-year DFS was not different between MTAP groups (Figure 4D; *p =* 0.230).

### 3.3. MTAP Cell Line Editing and Differential Gene Expression

To further underpin the *MTAP* biological role in gliomas, the U251 cell line, which expressed *MTAP*, was transfected with MTAP CRISPR/Cas9 KO to create the stable knockout clones (U251MTAP−/−). On the other hand, SW1088 cells that normally do not express *MTAP* were transduced by the MTAP human lentivirus to establish *MTAP* expression clones (SW1088MTAP+/+). Stable cell lines were confirmed by Western blotting post puromycin selection (Appendix A).

After *MTAP* gene editing, the transcriptomic profiles of U251MTAP−/− and SW1088MTAP+/+ cells as well as control cells (U251EV and SW1088LB) and parental cells (U251WT and SW1088WT) were evaluated for the expression of specific genes using the NanoString PanCancer Pathways Panel, which assessed thirteen canonical pathways (Notch, Wnt, Hedgehog, chromatin modification, transcriptional regulation, DNA damage control, transforming growth factor-beta (TGF-beta), mitogen-activated protein kinase (MAPK), The Janus kinase/signal transducers and activators of transcription (JAK/STAT), Phosphoinositide 3-kinase (PI3K), RAS, cell cycle, and apoptosis). In the U251 cells, we found that the U251MTAP−/− clone showed 17 differentially expressed genes in comparison with controls, with seven (*ANGPT1, NGFR, FN1, COL11A1, SPP1, RASGRF1,* and *MAPK10*) being associated with the PI3K/RAS/MAPK pathways, three (*TGFBR2, THBS1,* and *BMP5*) with the TGF-β pathway, two (*IDH2* and *ERBB2*) belonging to the Driver pathway, and five genes (*TNFSF10, SFRP1, PLAT, NOTCH1,* and *BRIP1*) associated with other pathways (Figure 5A, Appendix A). When analyzing only differentially expressed genes (*p*. adjusted <0.05; log_2_FC≥±2), we found six upregulated (*ANGPT1, FN1, NGFR, RASGRF1, THBS1*, and *TNFSF10*) and three downregulated genes (*BMP5, COL11A1*, and *MAPK10*) in the MTAP KO cells (Figure 5B).

We further performed the same approach in the SW1088 gene-edited cells. In the SW1088MTAP+/+ clone, we observed significant alterations in six genes (*IL6R, GPC4, IL13RA2, MAP2K6, E2F5,* and *ETV1*) belonging to JAK-STAT, PI3K, WNT, MAPK, transcriptional regulation, and apoptosis pathways (Figure 5C, Appendix A). Following adjustment (*p*-adjusted <0.05; Log_2_FC>≥2), only *GPC4* and *IL13RA2* were differentially expressed when compared with control cells (Figure 5D, Appendix A). Subsequently, an analysis of the functional connections among the proteins encoded by the 17 and the six genes differently expressed by U251MTAP−/− and SW1088MTAP+/+ was performed using STRING software. For U251MTAP−/− cell, 11 of 17 genes had at least two connections, as shown in Figure 5E. Notably, these 11 genes were centered on five genes (*NOTCH1, ANGPT1, ERBB2, THBS1,* and *FN1*). On the other hand, for SW1088MTAP+/+, only *IL6R* and *MAP2K6* showed the signal connection (Figure 5F).

### 3.4. MTAP Deletion Does Not Modulate Glioma Cell Proliferation

To address the functional impact on cell proliferation, U251MTAP−/−, U251EV, and U251WT (1x10^5^ cells) as well as SW1088MTAP+/+, SW1088LB, and SW1088WT (5 × 10*^3^* cells) were seeded in E-plates, and then the cell index and the doubling time were evaluated (Figure 6). We observed that U251MTAP−/− showed analogous kinetic trace characteristics when compared to U251EV after 72 h (Figure 6A). When we analyzed the cell doubling time, within 24 h (cell doubling time 2.36 ± 1.51 vs. 3.32 ± 0.88; *p* = 0.400) and persisting up to 72 h (cell doubling time; 10.36 ± 1.64 vs. 11.64 ± 1.40, *p* = 0.800), the U251MTAP−/− cell showed increased proliferation rates in relation to U251EV, although it did not show a significant difference (Figure 6B, 24 h: *p* > 0.999; 48 h: *p* > 0.999; 72 h: *p* = 0.547). Likewise, the SW1088MTAP+/+ cells did not exhibit distinct proliferation rates (Figure 6C, *p* > 0.05) and doubling time (Figure 6D, 24 h: *p =* 0.700; 48 h: *p =* 0.885; 72 h: *p =* 0.885) when compared with the control clone (SW1088LB). Together, these results suggest that the *MTAP* gene alone does not modulate cell proliferation.

### 3.5. Loss of MTAP Gene is Not Associated with Cell Migration and Invasion

To investigate whether the loss of *MTAP* could affect glioma cell motility and invasiveness, we used the transwell cell migration assay (Figure 7A,B). U251MTAP−/− cells promoted decreased ability of migration compared with U251EV by 10% (U251MTAP−/− 552.5 ± 205.3 vs. U251EV: 614.2 ± 223) (Figure 7A, *p* > 0.999), while the overexpressed clone SW1088MTAP+/+ showed an increase in migration ability about of 19% (SW1088LB: 557.8±40.8 vs. SW1088MTAP+/+: 687.6 ± 138.1) (Figure 7B, *p* = 0.400). Regarding the invasive properties of edited cells, as shown in Figure 7C, U251EV cells (601.7 ± 201.6) exhibited an 8% greater invasion compared to U251MTAP−/− (553.5 ± 92.1), yet it was not statistically significant (*p =* 0.857). Likewise, an increase of SW1088MTAP+/+ invasion was observed in ~12% (SW1088MTAP+/+: 264.3 ± 50.7 vs. SW1088LB: 230.7 ± 31.9) compared to SW1088LB cell, also not statistically significant (Figure 7D, *p =* 0.628). Moreover, we obtained similar results when comparing U251MTAP−/− and SW1088MTAP+/+ with U251WT and SW1088WT parental controls, respectively (data not shown).

Next, to support in vitro results, we evaluated the differential expression of 24 genes associated with invasion and migration processes. Only two of these genes were differently expressed in the U251 MTAP gene-edited cell line compared to U251EV cells (Appendix A). The *FN1* gene was found to be upregulated (log_2_FC = 2.26; *p =* 0.0006) and, inversely, *MAPK10* was found downregulated (log_2_FC = −3.28; *p* < 0.0001) in U251MTAP−/− compared to U251EV (Appendix A). Furthermore, no differential expression was found between SW1088MTAP+/+ and control (Appendix A).

## 4. Discussion

In the present study, we evaluated *MTAP* expression in a large series of gliomas. Moreover, by using a genome edition gain- or loss-of MTAP-function approach in glioma cell lines, we explored MTAP biological impact on gliomas. We observed that, despite the frequent loss of MTAP expression in high-grade gliomas, *MTAP* was not associated with a worse outcome, and the in vitro models showed that *MTAP* does not affect cell line proliferation, invasion, and migration.

Initially, using an in silico approach, we showed that 48% (168/350) of gliomas had 9p21 locus deleted, and this deletion was directly associated with reduced *MTAP* expression. In fact, Zhao and Zhao in a tumor suppressor pan-cancer study reported the association between copy number loss (9p21) and reduced *MTAP* mRNA expression [49]. In addition, the homozygous deletion of 9p21 locus was identified to be significantly associated with a decrease of *MTAP* (mRNA and protein) expression levels [50]. This region of the human genome is especially interesting because *MTAP* is located close to a fragile site [51]. The deletion of this region usually causes co-deletion of *MTAP* and some classic and well-known tumor suppressor genes such as *CDKN2A/B* [52]. The cyclin-dependent kinase inhibitor 2A (*CDKN2A*) and the cyclin-dependent kinase inhibitor 2B (*CDKN2B*) genes both encode putative regulators of cyclin-dependent kinases on chromosome 9p21 that could lead to uncontrolled cell proliferation and have been associated with poor prognosis in many cancer types included gliomas [23,53,54,55]. The 9p21 chromosomal region deletion is rare in grade I but frequent in grade IV glioma [39]. Our findings, in combination with the detection of other biomarkers, show *CDKN2A/B* could be used to predict 9p21 deletion and to stratify patients for stricter surveillance. By analyzing the TCGA-GBM dataset, we also observed that deletion, rather than *MTAP* promoter methylation, is associated with MTAP mRNA expression. Our results contrast with a recent study that found a significant association of MTAP methylation with gene expression [40]. However, Hansen et al. observed a very low coefficient of determination (R2) (0.19) with a low correlation coefficient (R) value (0.44) clearly below the values considered as strong correlation (>0.6); therefore, caution should be taken in interpreting the reported correlation between DNA promoter methylation and MTAP expression described [40].

We then evaluated *MTAP* expression profile in accordance with the molecular subtype of glioblastomas described by Verhaak et al. [56] and Noushmehr et al. [16]. When using the TCGA-GBM dataset, we observed that *MTAP* loss of expression was more pronounced within the classical subtype (65.2%), corroborating previous reports that patients with classical subtype are characterized by *CDKN2A* gene deletion that is contiguous with the *MTAP* gene [26]. On the other hand, only 7.4% of the G-CIMP glioblastomas showed loss of MTAP, which is associated with secondary glioblastoma, tumor-harboring mutation of the *IDH1* gene, and lesions that progressed from LGG [16,57,58].

In glioma cell lines, we observed loss of *MTAP* in high-grade glioma cells, contrasting with low-grade glioma (40% vs. 0%). Interestingly, there was no loss of *MTAP* expression in pediatric GBM cell lines, whereas 50% (4/8) of adult GBM cell lines presented the loss of expression. These findings are in line with the less frequent (10–19%) 9p21 locus deletion event in pediatric HGG cases [8,59].

We further characterized MTAP expression by IHC, and we sought to associate it with its prognostic value. We previously reported that MTAP loss occurs in less than 15% of pilocytic astrocytomas (WHO grade I) [42]. Herein, we extended the immunohistochemistry analysis to a large set of diffuse infiltrative astrocytomas and observed association between loss of MTAP expression and malignancy grade of gliomas. This result is in line with the report by Suzuki et al. [60] that identified the high frequency of *MTAP* deletion (60%) in glioblastoma series but rare presence in low-grade glioma [39,42]. Differently, in glioma cell lines, the loss of MTAP for pediatric glioblastoma (54.8%) was higher than in adult glioblastoma (45.6%) series. This finding was unexpected and may suggest that our series of pediatric GBM harbored histological characteristics such as those described in the receptor tyrosine kinases (RTK) II classic subgroup. These tumors exhibit features of copy number alterations of adult glioblastomas [8,61]. In a recent study, Frazão et al. [39] identified, in a series of pediatric gliomas, rare episodes of deletion in grade I glioma (12.2%) but frequent 9p21 deletion in high-grade glioma (62.5%). These results point to the correlation of *MTAP* deficiency with increased malignancy and histological subtype of gliomas [11,38,42,50,60].

Despite the frequent loss of expression associated with higher-grade lesions, when evaluating its clinical impact in our HGG and adult GBM cases, the loss of MTAP was surprisingly correlated with a better prognosis. One hypothesis that may explain this fact is that patients diagnosed with the classical subtype (frequent chromosome 9p deletion) presented better survival after intensive therapy (chemotherapy/radiotherapy) [56]. Our dataset is composed of patients with primary GBM characteristics (median age of 59 years, IDH1-mutated in 3.9%) with 79% (287/363) having undergone chemo or radiotherapy treatment, corroborating this result. In addition, through the bioinformatics approach, we did not observe any association of *MTAP* expression/deletion with overall survival (*p* = 0.942) and disease-free survival (*p* = 0.230) from the TCGA-GBM dataset. These results are not in accordance with Hansen et al. [40], which also analyzed the TCGA-GBM dataset and reported that MTAP-deleted cases presented worse disease-free survival (DFS) when compared to MTAP-normal patients. The lack of accuracy of DFS in GBM prognostic evolution, particularly outside a clinical trial assessment, is already known; therefore, overall survival (OS) is the most used criterion. When analyzing the same TCGA GBM dataset, we observed that MTAP-deleted patients did not show a worse OS when compared with MTAP-normal patients. Additionally, in agreement with Hansen et al. [40], we performed the DFS analysis of the same GBM dataset, considering the gold standard DFS endpoints (7), namely relapse and death events, different to the study by Hansen that only considered relapse and censured the events of death. The results also showed a lack of association between MTAP deletion and GBM patients’ prognosis.

Our NanoString Pan-Cancer Panel analysis of *MTAP* gene-edited cell lines showed a significantly altered expression of genes associated with RAS/MAPK/PI3K-AKT and apoptosis pathways after *MTAP* gene knockout. Signaling by these pathways governs fundamental physiological processes, such as cell proliferation, differentiation, metabolism, cell death, and survival [62,63,64]. Out of nine genes differentially expressed in our study, five are included in the PI3K-AKT pathway, with three of them—*FN1, NGFR,* and *THBS1*—being associated with tumor progression [65,66,67,68]. Besides, upregulated *ANGPT1* and downregulated *COl11A1* are related to a better prognosis in lung cancer metastasis and proliferation in colorectal cancer [68,69]. Another gene, *TNFSF10,* was identified as upregulated in U251MTAP−/−. This gene is frequently associated with the pro-apoptotic process by caspase 8 activation [70,71]. Curiously, only two genes, GPC4 and *IL13RA2*, showed altered expression for SW1088MTAP+/+ cell line when compared to the control. These two genes exhibit differential expression and controversial function in various tumors acting as tumor promoters and inhibitors in a cancer type-specific manner [72,73,74,75].

Importantly, in U251MTAP−/− and SW1088MTAP+/+ edited glioma cells, we showed that MTAP does not regulate glioma cell proliferation, migration, and invasion mechanisms, major cancer biological features of classic tumor suppressor genes. The results are consistent with the ones observed in our mRNA expression analysis obtained by the Pan-Cancer Pathway Panel assay. For these genes, significant differences were observed only for *FN1* and *MAPK10* between U251MTAP−/− and U251EV without difference for other genes evaluated. A recent study with prostate cancer cells silenced for *MTAP* also showed similar growth rates to controls [33]. Furthermore, it was observed in hepatocellular carcinoma that the re-expression of *MTAP* did not change the proliferation rates when compared to mock controls [28]. In gliomas, a recent study showed compelling data that MTAP loss is responsible for epigenetic remodeling and stemness properties [40]. The comparison of the present study with Hansen et al. [40] shows an important limitation related to culture systems. Our findings were obtained from 2D monolayer cultures, a high-serum adherent culture system that failed to maintain glioma stem-like subpopulation of cells and overrepresented only neoplastic cells [76]. We highlight that the process of in vitro culturing of GBM cells is fraught with challenges [77]. Although Hansen and colleagues provided the first evidence for relevance of MTAP loss in stemness-induction in GBM, their study also failed to account for a wide range of factors known to influence tumor growth, migration, invasion, and resistance to therapy. Therefore, for now, despite efforts, there is a lack of satisfactory methods for detailed monitoring of molecular and cellular determinants of tumor growth in patients as well as ideal in vitro systems for modeling of these processes [77]. Thus, additional studies with implementation of new culture models that better recapitulate the complex reality of glioma growing in situ are warranted to address this topic.

*MTAP* has been reported as an important therapeutic regulator. It may contribute to selective therapy with thiopurines in combination with MTA in *MTAP*-deleted tumors as a strategy to induce selective cell death [19,40,78]. However, in the context of treatment strategy, MTAP status as a factor of molecular vulnerability to increase the efficacy of treatment in patients with glioma should be further studied with caution. Our data provide arguments towards the lack of strong biological importance of MTAP in gliomagenesis. Clearly, a better understanding of the molecular changes of *MTAP*-depleted cells is needed to exploit this direct and indirect molecular vulnerability, increasing the efficacy of treatment in patients with glioma.

In summary, the present work showed that loss of *MTAP* expression is a frequent event in high-grade gliomas. In silico and in vitro models provided evidences towards the lack of strong biological importance of *MTAP* in gliomas. This study also showed that, despite the frequent loss of *MTAP*, it does not have a clinical impact in survival and does not act as a canonic tumor suppressor gene in gliomas.

## Figures and Tables

**Figure 1 cells-09-00492-f001:**
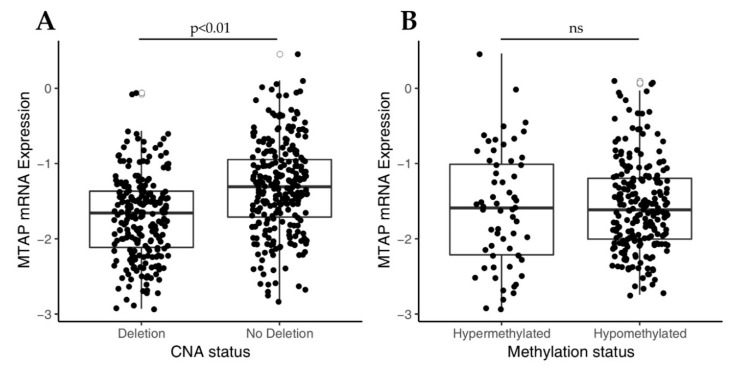
Analysis of *MTAP* mRNA expression in The Cancer Genome Atlas glioblastoma (TCGA-GBM) dataset correlates to copy number alterations (CNA) status for 9p21 locus and *MTAP* promoter methylation. (**A**) Box-plot analysis showing downregulation in *MTAP* expression in 9p21 homozygous deleted samples compared to matched normal samples (*p*-value < 0.01). (**B**) Box-plot of *MTAP* expression and promoter methylation in TGCA-GBM dataset. *ns*: not significant *p*-value.

**Figure 2 cells-09-00492-f002:**
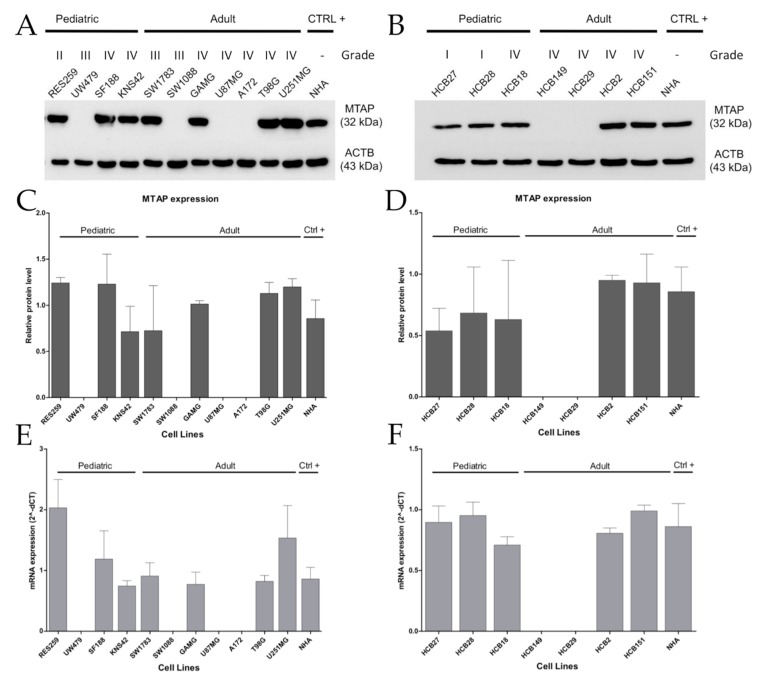
Loss of MTAP expression is more present in high-grade glioma (HGG) cell lines. MTAP protein expression analysis of established (**A**) and short-term primary glioma cells (**B**) assessed by Western blotting. Plot representative of MTAP protein (**C**,**D**) and gene (**E**,**F**) expression in glioma cell lines. Samples were normalized to *HPRT-1* (mRNA) and beta-actin (ACTB) (protein) as an endogenous control. Abbreviations: CTRL+: positive control.

**Figure 3 cells-09-00492-f003:**
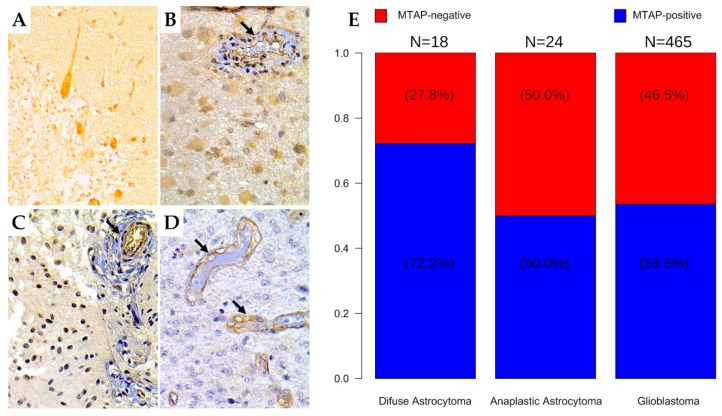
Immunohistochemistry staining for MTAP in human brain tumor tissues. The human brain tissue slide used for this study contained 507 cases of patients with different grades of gliomas in duplicates. Representative images from (**A**) normal cerebellum tissue (positive control); (400×) (**B**) diffuse astrocytoma (grade II); (400×) (**C**) anaplastic astrocytoma (grade III); 400× (**D**) glioblastoma (grade IV); 400×. Arrows indicate vessels with endothelial staining for MTAP protein (positive internal control). (**E**) Frequency of loss of MTAP protein expression in the glioma dataset according to histologic subtype.

**Figure 4 cells-09-00492-f004:**
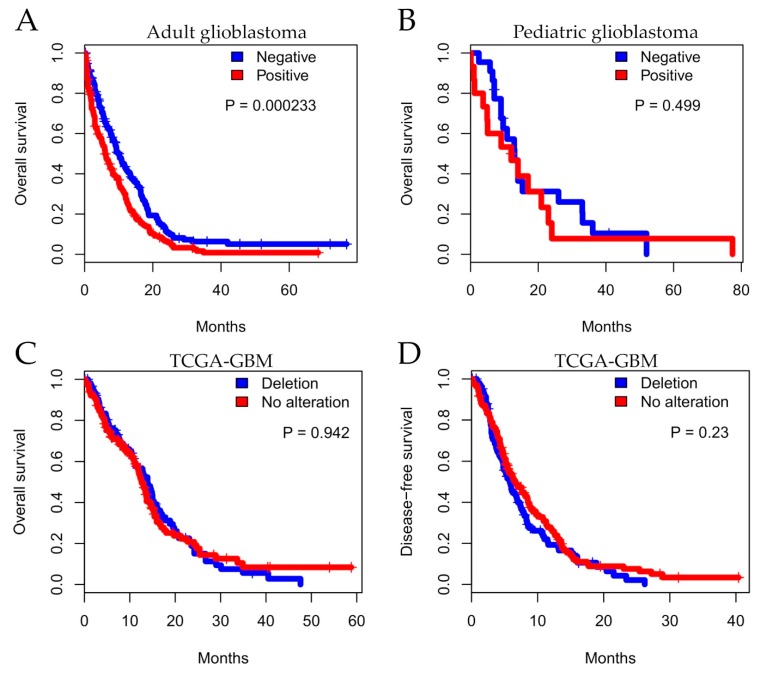
Kaplan–Meier analysis for *MTAP* expression associated with patient survival. Overall survival for a series of patients with glioma diagnosis according to histopathological grade. Shown for (**A**) adult glioblastoma, (**B**) pediatric glioblastoma, and (**C**) TCGA-GBM dataset status and (**D**) five-year disease-free survival (DFS) for TCGA-GBM dataset.

**Figure 5 cells-09-00492-f005:**
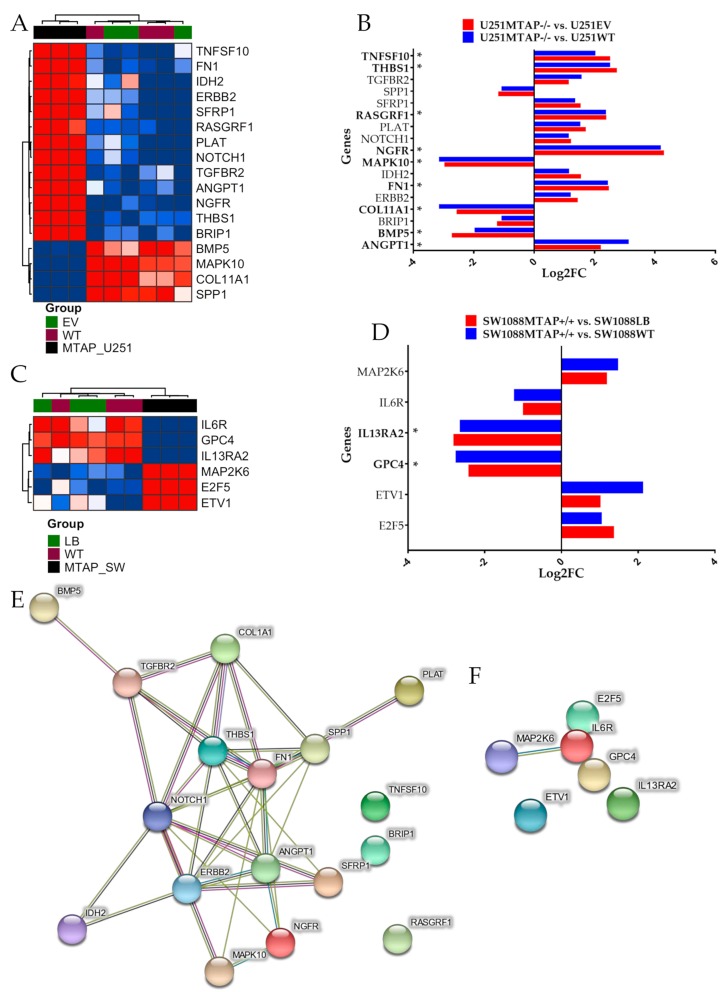
Detection of differentially expressed genes affected by *MTAP* gene-edited in glioma cell lines. (**A**) Heatmap representing the expression profile of the most differentially expressed genes for U251MTAP−/− cell compared to U251EV and U251WT cells. (**B**) Bar plots showing gene expression as the mean SD of log changes of U251MTAP−/− relative to U251EV and U251WT cells. (**C**) Heatmap representing the expression profile of the most differentially expressed genes for SW1088MTAP+/+ cell compared to SW1088LB and SW1088WT cells. (**D**) Bar plots showing the gene expression as the mean SD of log changes of SW1088MTAP+/+ relative to SW1088LB and SW1088WT cells. (**E**,**F**) Isolated networks of protein−protein interaction using STRING (http://www.string-db.org). The weight of these lines represents the confidence within which a predicted interaction occurs. Interactions networks for 17 and six proteins differentially regulated in U251MTAP−/− and SW1088MTAP+/+ cells compared to controls U251EV and SW1088LB, respectively. Rows represent genes and columns represent cell lines. Red pixels: upregulated genes; blue pixels: downregulated genes. The intensity of each color denotes the standardized ratio between each value and the average expression of each gene across all samples. Each sphere represents an individual protein, and edges represent protein–protein associations. A red line indicates the presence of fusion evidence; a green line indicates neighborhood evidence; a blue line indicates co-occurrence evidence; a purple line indicates experimental evidence; a yellow line indicates text-mining evidence; a light blue line indicates database evidence; and a black line indicates co-expression evidence. Asterisks (*) labeling the bar indicates a significant difference (*p*-value < 0.05 and log_2_FC≥±2) between the gene expression of respective groups and the control. The network nodes represent proteins. Edges represent protein–protein associations. Student t-test: * *p* < 0.05, ** *p* < 0.01, *** *p* < 0.001, *ns*: not significant

**Figure 6 cells-09-00492-f006:**
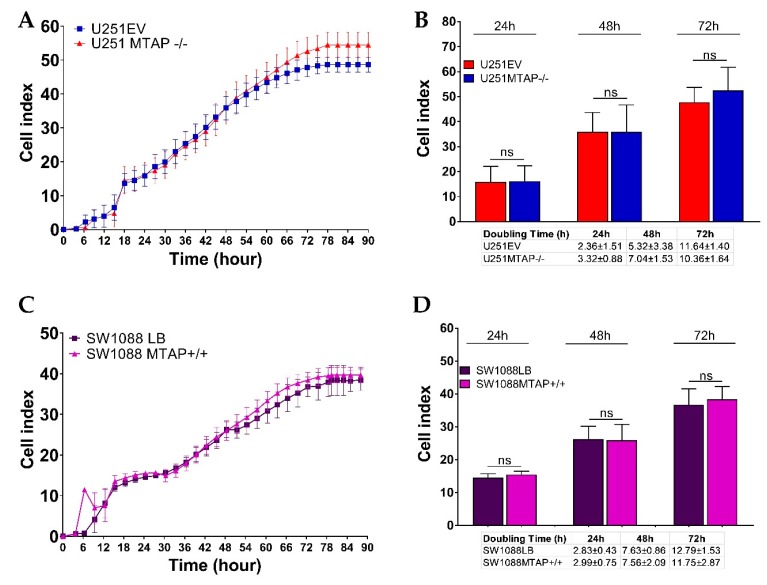
Dynamic monitoring of cell proliferation using impedance technology. Cell index values representative for U251MTAP−/− vs. U251EV (**A**) and SW1088MTAP+/+ vs. SW1088LB (**C**) measured by the Real Time Cell Analysis (RTCA) assay for 90 h. Plot representative of cell index and doubling time at 24, 48, and 72 h for U251MTAP−/− vs. U251EV (**B**) and SW1088MTAP+/+ vs. SW1088LB (**D**). Plot representative of three independent experiments. Student t-test: * *p* < 0.05, ** *p* < 0.01, *** *p* < 0.001, *ns*: not significant.

**Figure 7 cells-09-00492-f007:**
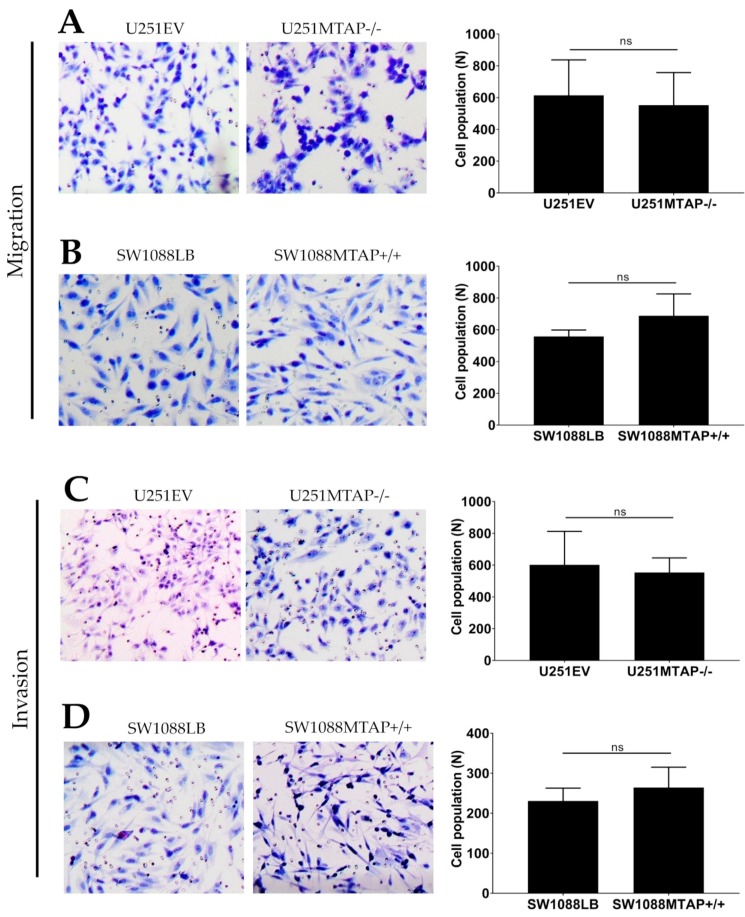
*MTAP* gene does not modulate migration and invasion abilities. Migration potential in both clone cell lines U251MTAP−/− (**A**) and SW1088MTAP+/+ (**B**) compared to controls (U251EV and SW1088LB) measured by Transwell assays and expressed as relative cell population. Transmembrane invasion assay of U251MTAP−/− (**C**) and SW1088MTAP+/+ (**D**) compared to controls (U251EV and SW1088LB). Quantification of migrated and invaded cells numbers per field. A minimum of 10 fields of two separate wells within each experiment were analyzed. The graphs represent data from three separate experiments. Student t-test: * *p* < 0.05, ** *p* < 0.01, *** *p* < 0.001, *ns*: not significant.

**Table 1 cells-09-00492-t001:** *MTAP* differential expression in subtypes of glioblastoma.

Subtype	*MTAP* Positive	*MTAP* Negative	Total	*p*-Value
Classical	23 (34.8%)	43 (65.2%)	66 (100%)	<0.001
Mesenchymal	36 (41.9%)	50 (58.1%)	86 (100%)	
Neural	19 (37.3%)	32 (62.7%)	51 (100%)	
Proneural	30 (49.2%)	31 (50.8%)	61 (100%)	
G-CIMP+	25 (92.6%)	2 (7.4%)	27 (100%)	

G-CIMP+: glioma-CpG island methylator phenotype.

**Table 2 cells-09-00492-t002:** Clinicopathological features of glioma patients and association with *MTAP* expression.

			MTAP Expression		
	Total	(%)	Negative (%)	Positive (%)	*p*-Value
**Gender (N = 491)** Male	300	(61.1)	141 (47.0)	159 (53.0)	0.503
Female	191	(38.9)	83 (43.5)	108 (56.5)	
**Age group (N = 504)**					
0–19	49	(9.7)	26 (53.1)	23 (46.9)	0.479
20–59	256	(50.8)	112 (43.8)	144 (56.2)	
>59	199	(39.5)	92 (46.2)	107 (53.8)	
**Location (N = 210)**					
Frontal Lobe	88	(41.9)	24 (27.3)	64 (72.7)	**0.013**
Parietal Lobe	41	(19.5)	9 (22.0)	32 (78.0)	
Temporal Lobe	66	(31.4)	28 (42.4)	38 (57.6)	
Occipital Lobe	10	(4.8)	5 (50.0)	5 (50.0)	
Cerebellum	5	(2.4)	4 (80.0)	1 (20.0)	
**KPS (N = 168)**					
<70	75	(44.6)	11 (14.7)	64 (85.3)	0.539
>70	93	(55.4)	18 (19.4)	75 (80.6)	
**Grade (N = 507)**					
Low Grade	18	(3.6)	5 (27.8)	13 (72.2)	0.149
High Grade	489	(96.4)	228 (46.6)	261 (53.4)	
**Histologic Subtype (N = 507)**					
DA (WHO Grade II, NOS)	18	(3.6)	5 (27.8)	13 (72.2)	0.272
AA (WHO Grade III, NOS)	24	(4.7)	12 (50.0)	12 (50.0)	
Ped. GBM (WHO Grade IV, NOS)	42	(8.3)	23 (54.8)	19 (45.2)	
Ad. GBM (WHO Grade IV, NOS)	423	(83.4)	193 (45.6)	230 (54.4)	
**GBM status mutation (N = 460)**					
***IDH1* mutation (N = 229)**					
No	220	(96.1)	117 (53.2)	103 (46.8)	0.185
Yes	9	(3.9)	7 (77.8)	2 (22.2)	
***TERT* promoter mutation (N = 45)**					
No	11	(24.4)	3 (27.3)	8 (72.7)	0.687
Yes	34	(75.6)	7 (20.6)	27 (79.4)	
***MGMT* promoter methylation (N = 186)**					
No	120	(64.5)	75 (62.5)	45 (37.5)	0.753
Yes	66	(35.5)	39 (59.1)	27 (40.9)	

Abbreviation: DA: diffuse astrocytoma, AA: anaplastic astrocytoma, Ped. GBM: pediatric glioblastoma, Ad. GBM: adult glioblastoma KPS: Karnofsky Performance Status, NOS: not otherwise specified. T-test (*p* < 0.05).

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
