# Peer review of "Loss of 5′-Methylthioadenosine Phosphorylase (MTAP) is Frequent in High-Grade Gliomas; Nevertheless, it is Not Associated with Higher Tumor Aggressiveness"

_cells, 2020, doi:10.3390/cells9020492_

Round 1

Reviewer 1 Report

The manuscript “Loss of 5’-methylthioadenosine phosphorylase (MTAP) is frequent in high-grade gliomas, nevertheless, it is not associated with higher tumor aggressiveness” is a thoughtful, well designed series of experiments. In this manuscript, the authors explore the expression of MTAP in glioma of different grades and evaluate the association of its expression with patient survival and markers of proliferation and invasion in vitro. The role of MTAP in high grade glioma (HGG) is relatively unstudied, and the conclusions of this research group are made more interesting by the fact that they conflict with a published report in 2019 by Hansen et al (ref. 33).

Despite this, there are some significant limitations of the manuscript. Most importantly, the authors should address the findings of Hansen et al. a bit more specifically, particularly in the discussion, when addressing areas where their findings diverge and potential explanations. The findings of the manuscript would be further strengthened by a few additional in vitro experiments as well.

The introduction does not clearly identify why MTAP is of interest to the authors, nor why they assume it could be a driver mutation capable of driving tumorigenesis. Throughout the manuscript this is a nagging concern. While the authors demonstrate that in their system MTAP loss doesn’t lead to increased growth rate or migration, this may be due to their use of a high-serum adherent culture system that likely does not have any D133+stem cells. As this is the population MTAP may increase, the culture system may not allow for an effect of MTAP loss to be readily observed. The idea behind overexpression and knockout of MTAP in cell lines is good. These experiments should be reproduced for a larger number of cell lines (ideally 2+ for GOF and LOF, each). It would also be preferable to perform these experiments in neurosphere culture system, where cell-cell interactions are maintained in a fashion that preserves the stem-cell niche. One potential explanation for the lack of effect from MTAP loss on migration/growth may be the fact that its effect appears to come through increasing the CD133+ stem cell population within the tumor (Hansen et al.) and this effect may be lost in the authors’ culture system. It’s not clear that 9p21 loss is a good surrogate only for MTAP loss as CDKN2A/B are also located there. This should be mentioned in the discussion as a potential caveat.   Minor Points: The manuscript would benefit from proofreading by a native English speaker. In some places, the meaning of the authors’ writing becomes a bit unclear secondary to the grammar. In several places in the text the authors overstate a non-significant finding. Specifically, lines 454-457 “8% greater invasion …. Yet not statistically significant (n=0.857). Table 2, please correct for multiple comparisons when ascertaining significance. Figure 5 is unreadable. The genes listed in the text and in tables S2 and S3 don’t quite line up. Specifically, IL6R is listed under both in the text but is missing from table S2. The authors compare loss of MTAP expression across grades and show (possibly) lower levels of MTAP expression in HGG. The size of the LGG cohort is only 18 patients, however, making this type of analysis rather speculative. The authors pay undue attention to this seeming association in lines 323-325 and this should be toned down somewhat. Figure 4 – The reviewer appreciates that the authors have shown both OS and DFS for the TCGA data in figure 4.

Reviewer 2 Report

Menenzes and colleagues have studied the role of MTAP in glioma pathogenesis and have looked whether the loss of MTAP is associated with tumor aggressiveness. In a big series of different glioma types at different stages they could not find a correlation between MTAP loss and clinicopathological features of affected patients. In glioma cell lines with and without MTAP expression, no significant influence of MTAP presence/absence on proliferation, migration and invasion characteristics could be observed although loss or presence of MTAP affected the expression of components of pathways associated with proliferation/migration/invasion. Thus although the authors could not show an association between MTAP loss/presence and glioma aggressiveness and the paper may thus be of limited clinical relevance, it is interesting from a scientific point of view.

I have only a minor comment:

As the authors describe in materials and methods the analysis of IDH1 and TERT mutations as well as MGMT promoter methylation, a brief note about the role of these components for glioma pathogenesis (either in the introduction or materials/method section) would be helpful for those readers that are not deeply involved in glioma research.

Round 2

Reviewer 1 Report

The authors have done a thorough job responding to my concerns and clarifying the limitations of this study. It will be a solid addition to the scientific literature and increase our understanding of MTAP in glioma